# The Investigation of Copolymer Composition Sequence on Non-Isothermal Crystallization Kinetics of Bio-Based Polyamide 56/512

**DOI:** 10.3390/polym15102345

**Published:** 2023-05-17

**Authors:** Diansong Gan, Yuejun Liu, Tianhui Hu, Shuhong Fan, Xiaochao Liu, Lingna Cui, Ling Yang, Yincai Wu, Lily Chen, Zhixiang Mo

**Affiliations:** 1Key Laboratory of Advanced Packaging Materials and Technology of Hunan Province, School of Packaging and Materials Engineering, Hunan University of Technology, Zhuzhou 412007, China; gands@zztep.cn (D.G.); f1s1h1@163.com (S.F.); xcliu_2014@163.com (X.L.); lncui1102@126.com (L.C.); xmwuyincai@fjirsm.ac.cn (Y.W.); lily_jasmine_001@163.com (L.C.); mzx13873355623@163.com (Z.M.); 2Zhuzhou Times Engineering Plastics Industrial Co., Ltd., Zhuzhou 412008, China; huth@zztep.cn

**Keywords:** copolymer PA56/512, bio-based polymer, thermal properties, non-isothermal crystallization kinetics

## Abstract

A new bio-based polyamide 56/512 (PA56/512) has been synthesized with a higher bio-based composition compared to industrialized bio-based PA56, which is considered a lower carbon emission bio-based nylon. In this paper, the one-step approach of copolymerizing PA56 units with PA512 units using melt polymerization has been investigated. The structure of the copolymer PA56/512 was characterized using Fourier-transform infrared spectroscopy (FTIR) and Proton nuclear magnetic resonance (^1^H NMR). Other measurement methods, including relative viscosity tests, amine end group measurement, thermogravimetric analysis (TGA) and differential scanning calorimetry (DSC), were used to analyze the physical and thermal properties of the PA56/512. Furthermore, the non-isothermal crystallization behaviors of PA56/512 have been investigated with the analytical model of Mo’s method and the Kissinger method. The melting point of copolymer PA56/512 exhibited a eutectic point at 60 mol% of 512 corresponding to the typical isodimorphism behavior, and the crystallization ability of PA56/512 also displayed a similar tendency.

## 1. Introduction

Polyamides (PAs) have displayed versatile potential in the fabrication of aluminum soft packaging film, especially for lithium-polymer batteries. PA also offers advantages such as high thermal stability, excellent chemical resistance and superior mechanical performance due to its unique amide group structures and arrangement of hydrogen bonds [1,2,3]. The aluminum–plastic film is a key part of lithium-polymer batteries, contributing to the safety, long-term stability and whole service life time of the battery itself [4]. The biaxially oriented polyamide (BOPA) film made of PA6 is commonly used as the outer layer of aluminum–plastic film. It requires good impact resistance, puncture resistance, insulation performance and friction resistance to protect the inner layer structure of the battery during normal usage [5]. With the growth of the NEV market, the use of eco-friendly polyamide films has been a hot topic, in line with the new global low-carbon emission strategy and trend.

The cost-effective industrial production of bio-based 1,5-pentanediamine from Cathay Biotech Inc. has attracted increased research for the development of bio-based polyamides. The bio-based 1,5-pentanediamine is obtained by the decarboxylation of lysine produced by the biological fermentation of natural raw materials such as starch and straw [6,7]. Bio-based polyamides with the required characteristics and structure, such as long-chain polyamides, high-temperature polyamides and transparent polyamides, can be produced using polycondensation processes with dioic acids of different chain lengths [8,9,10,11,12,13]. The commercialized PA56, polymerized from bio-based 1,5-pentanediamine and petroleum-based adipic acid, is a promising matrix film resin for use in the next generation of aluminum soft packaging film with green chemical energy applications in lithium–polymer batteries.

The low toughness and high water absorption capacity of pure PA56 would negatively affect the mechanical properties, electrical properties and dimensional stability of the products. To address this issue, a compounding modification and copolymerization process has often been used for the improvement of the performance of PA56 [14,15]. Recent work by Xu et al. [16] has demonstrated that the blend of PA56 and EPDM-g-MAH can significantly improve the toughness and heat resistance of the system. Yan et al. [17] proposed that PA56 could be used as the internal lubricating and nucleating agent in a blend system of PA56 and polyethylene terephthalate. Yang et al. [18] reported that the inherently flame-retardant PA56 copolymers (FRPA56s) synthesized by melt polycondensation with monomer 1,5-pentanediamine (PDA), adipic acid (AA) and 9,10-dihydro-10- [2,3-di(hydroxy carbonyl)propyl]-10-phosphaphenanthrene-10-oxide (DDP) presented an improvement in the flame retarding performance. Studies on the synthesis and performance of copolymer PA5T/56 [19], PA10T/56 [20] and PA56/66 [21] have also been reported before, although there are still few studies on the modification of PA56 through the copolymerization of long carbon chain aliphatic polyamide monomers. The increasing length of the methylene segment of the copolymer polyamide would increase the possibilities for hydrogen bonding arrangements, conformational transitions and structure changes in the system [22].

This paper demonstrates a high-efficiency approach to copolymerizing PA56 units with PA512 units through melt polymerization. The relative viscosity and the amine end group values of the samples were measured using an Ubbelohde viscometer and Metrohm 888 Titrando, respectively. The structure of the synthesized PA56/512 was characterized by FTIR and ^1^H NMR, and the thermal properties of PA56/512 were measured by thermogravimetric analysis (TGA) and differential scanning calorimetry (DSC). The long carbon chain of PA512 allows its methylene group to undergo more flexible conformational transition in the co-polymer component, resulting in different non-isothermal crystallization behaviors. Furthermore, the non-isothermal crystallization behaviors of PA56/512 with different composition ratios at different cooling rates were studied based on the DSC data. Mo’s method and the Kissinger method were used to determine the crystallization behavior of PA56/512. In summary, this study presents a comprehensive analysis of the copolymerization of PA56 units with PA512 units through melt polymerization. The results provide valuable insights into the thermal and crystallization properties of the synthesized PA56/512 copolymer, which can be useful in various industrial applications.

## 2. Experimental Section

### 2.1. Materials

1,5-pentanediamine was purchased from Cathay Biotech Inc. (Shanghai, China). Adipic acid was purchased from Sinopec Yangzi Petrochemical Co. Ltd. (Jiangsu, China). Dodecanedioic acid, concentrated sulfuric acid (96%), trifluoroethanol and trifluoroacetic acid-d (isotopic) were purchased from Aladdin Industrial Corporation (Shanghai, China). All reagents used were of analytically pure quality. All solutions were prepared using deionized water.

### 2.2. Preparation of Copolymer PA56/512

#### 2.2.1. Neutralization Reaction

The adipic acid and dodecanedioic acid, with a total weight of 2.5 kg, were prepared in a 50% mixed aqueous solution according to the set molar ratio (8:2, 7:3, 6:4, 5:5, 4:6, 3:7, 2:8). Then, 1,5-pentanediamine was added dropwise to the above mixed aqueous solution under vigorous stirring at 80 °C. A transparent aqueous solution was obtained when the pH of the whole solution was regulated at 7.8.

#### 2.2.2. Polymerization Reaction

The above solution was transferred to a 10 L autoclave, and the air in the autoclave was replaced with nitrogen 3–5 times. The reactor was heated to 155 °C to remove 1.1 kg of water; the temperature was increased to 200 °C and the pressure was maintained at 1.5 MPa for an hour. After the autoclave was heated to 260 °C with 1.5 MPa pressure, the vapor in the autoclave was gradually released to reduce the pressure to normal. The system was vacuumed and reacted at −0.08 MPa for 1 h. Finally, the PA56/512 resin was obtained after the discharging, cooling and granulating process.

Besides this, pure PA56 and pure PA512 resins were prepared in the same way, named sample #1 and #9, respectively. The copolymers PA56/512 were prepared from monomeric adipic acid and dodecanedioic acid with ratios of 8:2, 7:3, 6:4, 5:5, 4:6, 3:7 and 2:8, named samples #2, #3, #4, #5, #6, #7 and #8, respectively. Figure 1 presents a schematic diagram of the polymerization process of PA56/512, as well as the structure of pure PA56 and pure PA512.

### 2.3. Characterization of Copolymer PA56/512

Fourier transform infrared spectrometry (FTIR, PerkinElmer Frontier, Waltham, MA, USA) was carried out to obtain the functional group information of copolymer PA56/512. ^1^H NMR was measured using a Bruker (Billerica, MA, USA) AVIII 500HD (500 MHz) instrument with a solvent of trifluoroacetic acid to analyze the chemical structures of samples. The relative viscosities of the samples with a concentration of 0.5 g/mL were measured using an Ubbelohde viscometer in sulfuric acid solution (96%) at 25 °C. The concentrations of the amine end group were determined using Metrohm 888 Titrando (Herisau, Switzerland). In total, 0.1 g PA56/512 was fully dissolved in 50 mL trifluoroethanol solution, which was titrated using the hydrochloric acid–ethanol solution to measure the concentrations of amine end groups. Thermogravimetric analyses (TGA/DSC1/1100SF, METTLER TOLEDO, Columbus, OH, USA) of the samples were performed in N_2_ flow with a heating increase of 10 °C/min from 25 °C to 700 °C. The thermal performance and the non-isothermal crystallization kinetics of the prepared samples were measured by differential scanning calorimetry (DSC 214, NETZSCH, Selb, Germany). All samples were heated from 25 °C to 280 °C, with a heating rate of 10 °C/min. After holding at 280 °C for 3 min, the program was set to cool down to 25 °C at the set cooling rates of 2 °C/min, 5 °C/min, 10 °C/min and 20 °C/min respectively. After holding at 25 °C for 3 min, the second heating program was carried out in the same way as the first. All the DSC tests above were conducted in a nitrogen atmosphere.

## 3. Results and Discussion

### 3.1. Structural Characterization of PA56/512

Figure 2 shows the FTIR spectra of PA56/512 with different composition ratios. The similar peak areas obeserved in samples #1–9 demonstrate that all the samples contained the same functional group. The stretching vibrations of N–H bonds can be seen at the wavenumber of 3300 cm^−1^. The characteristic peaks observed at 2927 cm^−1^ and 2857 cm^−1^ correspond to the asymmetric stretching vibration and symmetric stretching vibration of −CH_2_, respectively. The sharp peak overserved at 1636 cm^−1^ belonging to Amide Ⅰ resulted from the C=O stretching vibration. The two peaks in Amide Ⅱ and Amide Ⅲ, corresponding to the wavenumbers of 1540 cm^−1^ and 1270 cm^−1^ respectively, are attributed to the mixing interaction of N–H bending vibrations and C–N stretching vibrations. Additionally, the 930 cm^−1^ (Amide Ⅳ) and 690 cm^−1^ (Amide Ⅴ) peaks represent the C–C=O stretching vibration and N–H out-of-plane wagging vibration, respectively [23]. The presence of Amide Ⅰ–Ⅴ peaks indicates that all the PA56/512 samples (#1–9) prepared have a typical polyamide structure.

The ^1^H NMR spectra of PA56/512 with different composition ratios are shown in Figure 3. Regions #a (3.53–3.35 ppm) and #b (2.68–2.56 ppm) represent the characteristic peak positions of α–H (α_2_, α_4_) of diamines and α–H (α_1_, α_3_) of diacids in PA56/512, respectively. Likewise, all the β–H (β_1_, β_2_, β_3_, β_4_) of diamines and diacids in PA56/512 are displayed in region #c (1.85–1.56 ppm), while the remaining γ/δ/ε–H (γ_2_, γ_3_, γ_4_, δ_3_, ε_3_) appear in region #d (1.45–1.21 ppm). Furthermore, the ratios of integral data in each region of the ^1^H NMR spectra are listed in Table 1, including the theoretical and experimental data. It is evident that the experimental values are consistent with the theoretical value, indicating the successful preparation of PA56/512 with different composition ratios. Furthermore, the relative viscosities and amine end group values of PA56/512 are also listed in Table 1. It can be clearly seen that the relative viscosity and *M*_n_ values of PA56/512 for all ratios are above 2.4 and 19,000 respectively, revealing effective one-step polymerization in the copolymerization of PA56 and PA512.

### 3.2. Thermal Properties of PA56/512

Figure 4a,b illustrate the TGA and DTG variation curves of PA56/512 with different composition ratios used to measure the thermal stability of all the samples. The relevant data on the process of thermal decomposition are listed in Table 2. It is clear that pure PA56 and PA512, and their copolymers, exhibit the same one-step thermal decomposition process, which is consistent with the typical thermal degradation mechanism of polyamides [24,25]. In general, C–N bond breakage occurs first, accompanied by the cyclization of pentanediamine, the hydrolysis of chain-segments and other reactions, and then C–C bond breakage occurs at high temperature [26]. According to the thermal decomposition data shown in Table 2, the initial decomposition temperature (*T*_D_), 20% weight decomposition temperature (*T*_20%_) and maximum decomposition rate temperature (*T*_M_) mostly increased with the increase in 512 content in PA56/512, revealing the better thermal stability of pure PA512 compared to pure PA56, resulting from the lower hydrogen bond density of PA512 than that of PA56 [27].

The non-isothermal crystallization curves of PA56/512 with different composition ratios at different cooling rates are shown in Figure 5. It can be observed that the crystallization temperature of PA56/512 display a trend of first decreasing and then increasing with the increase in 512 content, accompanied by a peak shape of first widening and then narrowing, due to the crystallization competition between PA56 and PA512. Furthermore, the pattern of the above trend is consistent at different cooling rates, and the second heating cycle DSC curves of PA56/512 with different composition ratios at different cooling rates are shown in Figure 6. It can be clearly seen that the moving trend of the melting curve is analogous to that of the crystallization curve. Figure 6 and Figure 7 depict the melting point curves of PA56/512 with different contents of 512. The copolymerization system of PA56/512 has the lowest melting point with 40% 56 and 60% 512, corresponding to sample #6, which exhibits typical isodimorphism behavior [28]. The lowest melting point is regarded as a “eutectic” point [29,30], which is considered as a demarcation point for crystallization behavior dominated by homopolymer crystal types of 56 or 512. Therefore, the preparation of random copolymer PA56/512 with different composition ratios is a simple and efficient means to obtain a rearranged sequence structure with tunable thermal properties. Moreover, the investigated system exhibits a wide range of tunable Tm, enabling a broad range of processing temperatures to be achieved, which could meet different industrial application requirements.

### 3.3. Non-Isothermal Crystallization Kinetics of PA56/512

Based on the crystallization curves of PA56/512 with different composition ratios, five sets of samples, #1, #2, #3, #8, and #9, are selected for further non-isothermal crystallization kinetics analysis. Among them, the crystallization behaviors of #1–3 and #8–9 are dominated by 56 and 512, respectively. Figure 8 illustrates the non-isothermal crystallization curves of PA56/512 at different cooling rates. It can be seen that as the cooling rate increases, the crystallization exothermic peaks of each sample shift to the lower temperatures, along with a gradually widening peak. This result is mainly attributed to the time difference between the rearrangement and stacking of polymer chain segments during the crystallization process [31,32]. Consequently, the greater the cooling rate, the more obvious the crystallization hysteresis. Otherwise, with a high cooling rate, the temperature of the samples decreases more significantly within the same cooling time, while the activity of the polymer molecular chains is poor at low temperatures, resulting in decreased crystallization perfection and a wider crystallization peak shape.

The relative crystallinity *X(T)* of PA56/512 can be calculated according to the following formula [33]:(1)$$X\left(T\right)=\frac{\int_{T_{0}}^{T}\left(\frac{dH_{c}}{dT}\right)dT}{\int_{T_{0}}^{T_{e}}\left(\frac{dH_{c}}{dT}\right)dT}$$
where *T*_0_ and *T_e_* are the crystallization initiation temperature and crystallization termination temperature, respectively. d*H_c_*/d*T* refers to the heat flow at a certain temperature. Figure 9 shows the *X(T)*−*T* curve of the PA56/512 obtained from Formula (1). The relevant non-isothermal crystallization kinetic parameters of PA56/512 with different composition ratios are listed in Table 3. The parameter Φ refers to cooling rates; *T_p_* represents crystallization peak temperature; Δ*T_c_* refers to the temperature interval between *T*_0_ and *T_e_*; Δ*H_c_* is the enthalpy of crystallization; *t*_1/2_ refers to the semi-crystalline period. According to Figure 9 and Table 3, at the same cooling rate, the crystallization temperature ranges of samples #1 to #3 increase gradually, while those of samples #8 to #9 decreases gradually, which corresponds to the crystallization times of samples #1 to #3 (#9 to #8) gradually becoming longer, and the crystallization rate gradually decreasing.

The conversion relationship between crystallization time (*t*) and crystallization temperature (*T*) is shown in Equation (2) [32]:(2)$$t=\frac{\left|T_{0}-T\right|}{\Phi }$$

Figure 10 presents the *X(t)*−*t* curve of the PA56/512 obtained from Formulas (1) and (2). Combined with the results for the same sample in Figure 9 and Figure 10, it can be noted that the crystallization rate of copolymer PA56/512 initially increases and then decreases at a specific cooling rate. The curve is relatively flat in the early and late crystallization stages, with a lower crystallization rate. This is mainly attributed to the fact that the nucleation process dominated by the nucleation rate requires a large degree of undercooling in the initial stage of crystallization, resulting in a lower nucleation rate [34]. In the middle of crystallization, both the nucleation rate and crystal growth rate need to be considered, resulting in the maximum crystallization rate when they match each other. Although the nucleation rate remains high in the late crystallization stage, the crystallization rate is dominated by the crystal growth rate [35]. The low temperature in this stage causes the movement of polymer chain segments to be blocked, thereby reducing the crystal growth rate. According to the data in Figure 10 and Table 3, the semi-crystallization time *t*_1/2_ is ranked as #1 < #2 < #3 < #9 < #8, corroborating the above conclusion that the addition of 56 or 512 copolymerization components to the system will result in a lower crystallization rate than the original system (pure PA56 or PA512), similarly to the trend seen for melting point.

Mo et al. [36] combined the Avrami equation and the Ozawa equation to obtain a corresponding relationship with the same time and the same crystallinity. The equation is as follows [37]:(3)$$\mathrm{lg}\Phi =\mathrm{lg}F\left(T\right)-a\mathrm{lg}t$$
where the parameter *a* refers to the ratio of the Avrami index and the Ozawa index. The physical meaning of *F(T)* can be interpreted as the cooling rate value that must be selected for a certain polymer system to reach a certain relative crystallinity per unit time [38]. Thus, the *F(T)* value is inversely proportional to the crystallization rate.

Based on Formula (3), the lgΦ−lg*t* curves of the PA56/512 with different composition ratios are shown in Figure 11. The high fitting degree of the equation to the curve reveals its applicability to the analysis of non-isothermal processes related to the PA56/512 system. The relevant calculation parameters are listed in Table 4. The value *a* of the same sample is approximately constant, and undergoes no change with differences in its own *X(t)*, confirming a certain proportional relationship between the Avrami index and Ozawa index for a specific system. The *F(T)* of the same sample is proportional to the relative crystallinity, indicating that the larger the relative crystallinity, the smaller the average crystallization rate in this range. Comparing the values of *F(T)* with different copolymer ratio systems with the same crystallinity, such as *X(t)* = 20%, the values of *F(T)* rank as follows: 21.18 (#1) < 22.32 (#2) < 41.45 (#3) < 11.47 (#9) < 23.56 (#8). The average crystallization rate exhibits an inverse relationship, which is consistent with the previous analysis. In addition, it can be seen that the average crystallization rate of pure PA512 is better than that of pure PA56.

The activation energy of the non-isothermal crystallization of a polymer is often calculated using the equation of Kissinger, as follows [39]:(4)$$\frac{d\;\left[\ln\frac{\Phi }{T_{P}{}^{2}}\right]}{d\left(\frac{1}{T_{p}}\right)}=-\frac{\Delta E}{R}$$
where *T_p_* represents the crystallization peak temperature, Δ*E* refers to the activation energy, and R is the gas constant. The slope −Δ*E*/R can be obtained from the linear relationship between ln(Φ/*T_p_*^2^) and *T_p_*^−1^, and then the Δ*E* value can be calculated. The ln(Φ*/T_p_*^2^)−*T_p_*^−1^ curves of PA56/512 for samples #1, #2, #3, #8 and #9 are shown in Figure 12. The calculated crystallization activation energy data are listed in Table 5, along with the following rankings: #1 > #2 > #3 > #9 > #8 > #9 > #1. A high crystallization activation energy means high crystallization exotherm, accompanied by a strong molecular chain movement ability, a high crystallization rate and a strong crystallization ability [40]. Accordingly, pure PA512 has the best crystallization ability, followed by pure PA56. It can be inferred that the crystallization ability of copolymer PA56/512 is similar to the trend of its melting temperature, with the lowest level at the eutectic point, which conforms to all previous analyses of non-isothermal crystallization kinetics.

In general, the reconfigurable chain structure and tunable thermal properties of random copolymer PA56/512 can be realized by regulating the ratio of copolymerized components, resulting in a difference in the arrangement of hydrogen bonds, which affects the crystallization process of the system, accompanied by a change in crystallization kinetics. Furthermore, all studies demonstrate that the average crystallization rate of pure PA512 is higher than that of PA56, owing to the longer saturated carbon chain of PA512, the greater mobility of the long polymethylene segments, and the more flexible orientation of hydrogen bonds [41,42].

## 4. Conclusions

In this paper, a bio-based copolymer PA56/512 with different composition ratios was successfully synthesized by a one-step melt polymerization process. All the samples exhibited high relative viscosity and molecular weight, which makes them suitable for use in industrial application processes. The FTIR and ^1^H NMR spectra show that the actual structure of the prepared PA56/512 is consistent with the feed structure. TGA tests have indicated that the thermal stability of copolymer PA56/512 increased with an increasing proportion of 512. The second heating DSC curves of PA56/512 with different composition ratios reveal the typical isodimorphism behavior of the PA56/512, the lowest melting points of which are attained with 40% 56 and 60% 512 (sample #6). The non-isothermal crystallization kinetics analysis of PA56/512, undertaken using Mo’s method and Kissinger method, indicate that the crystallization ability of copolymer PA56/512 follows a similar trend to its melting temperature, with first decreasing and then increasing. The crystallization ability of pure PA512 was higher than that of PA56, which can be attributed to the high mobility and fast conformational transition of long polymethylene segments in the 512 component.

## Figures and Tables

**Figure 1 polymers-15-02345-f001:**
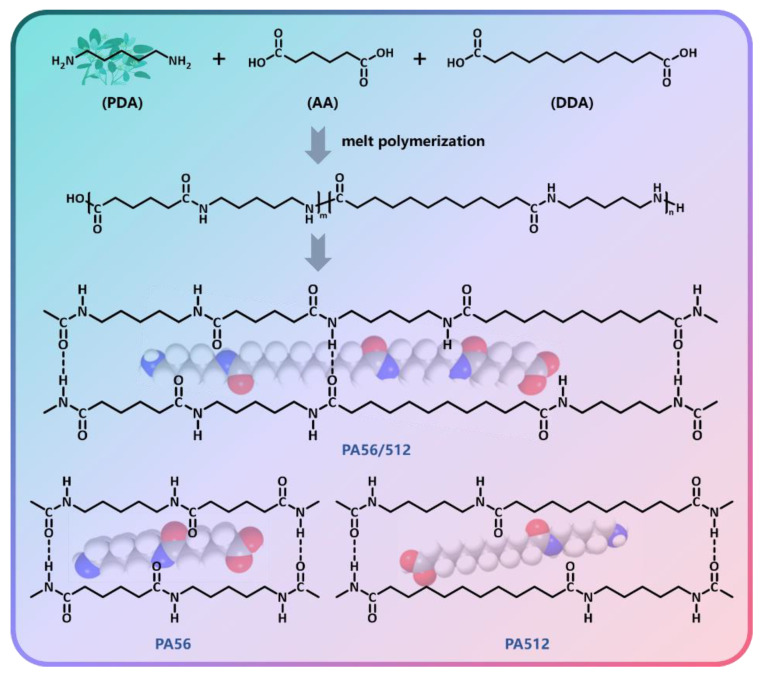
Schematic diagram of the polymerization process of PA56/512.

**Figure 2 polymers-15-02345-f002:**
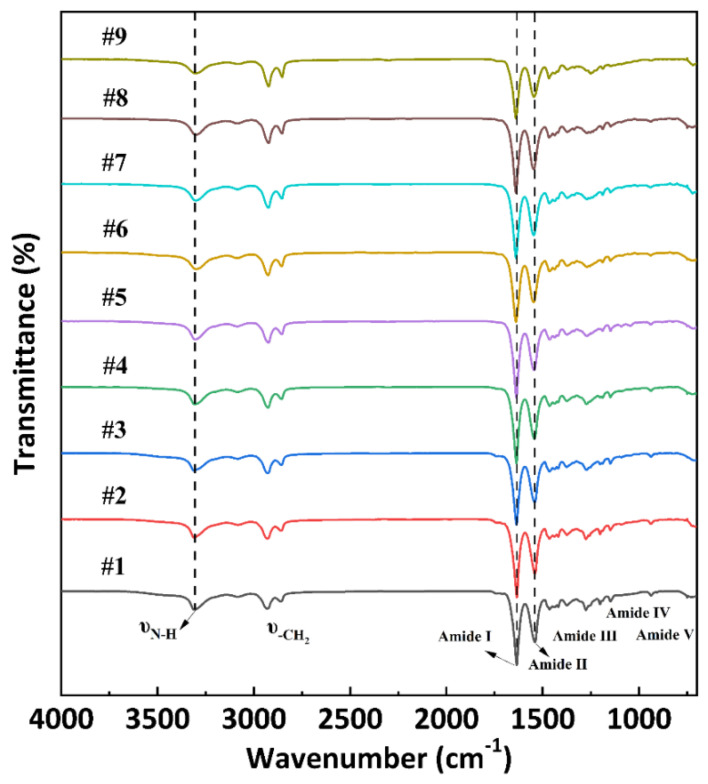
FTIR spectra of PA56/512 with different composition ratios.

**Figure 3 polymers-15-02345-f003:**
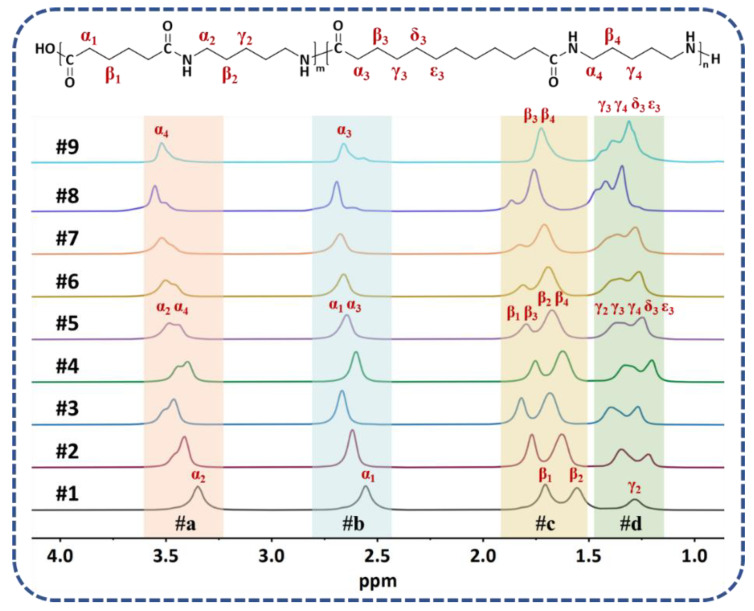
^1^H NMR spectra of PA56/512 with different composition ratios.

**Figure 4 polymers-15-02345-f004:**
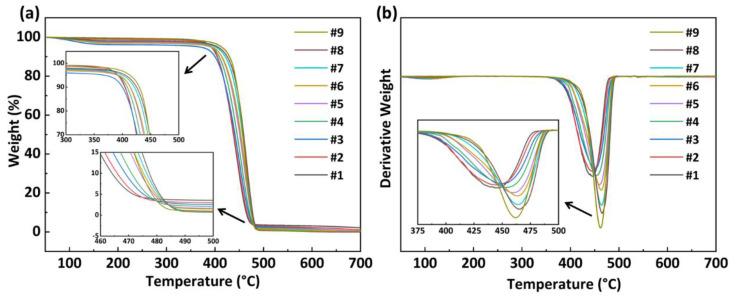
(**a**) TGA and (**b**) DTG variation curves of PA56/512 with different composition ratios.

**Figure 5 polymers-15-02345-f005:**
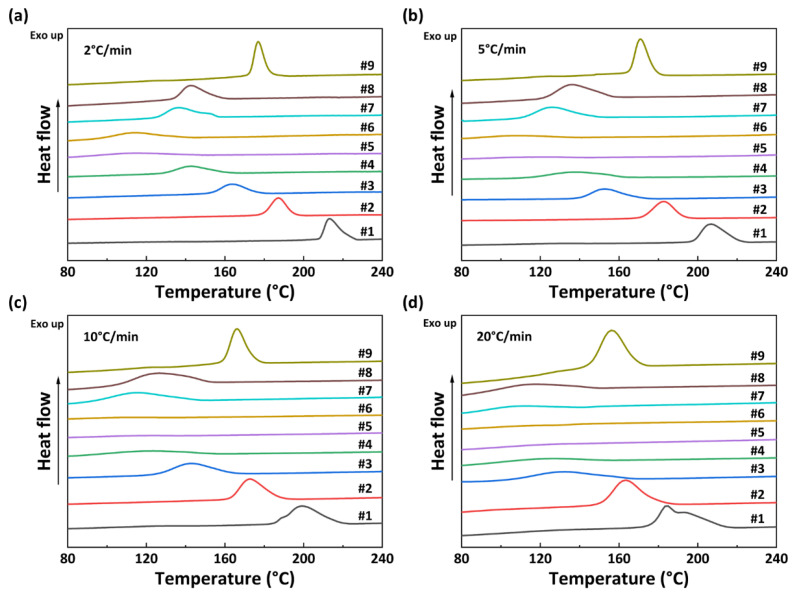
Non-isothermal crystallization curves of PA56/512 with different composition ratios at different cooling rates: (**a**) 2 °C/min, (**b**) 5 °C/min, (**c**) 10 °C/min, (**d**) 20 °C/min.

**Figure 6 polymers-15-02345-f006:**
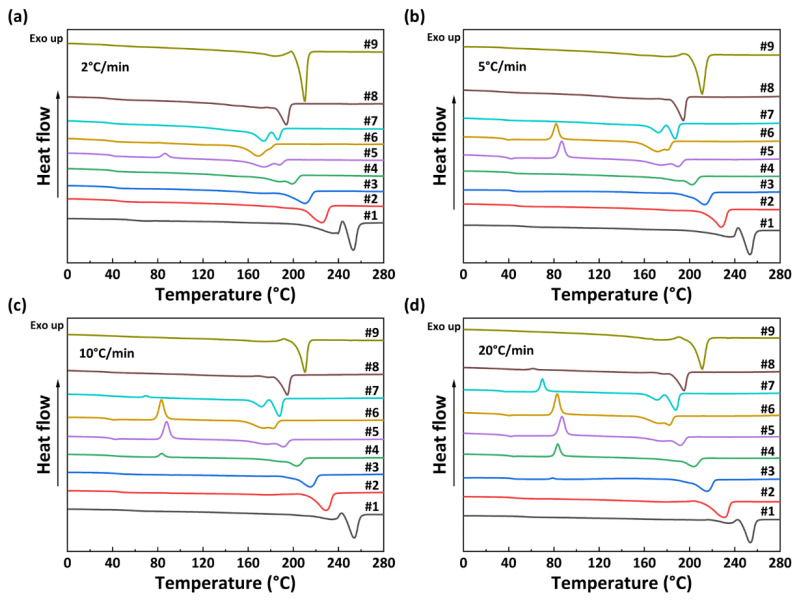
The second heating DSC curves of PA56/512 with different composition ratios at different cooling rates: (**a**) 2 °C/min, (**b**) 5 °C/min, (**c**) 10 °C/min, (**d**) 20 °C/min.

**Figure 7 polymers-15-02345-f007:**
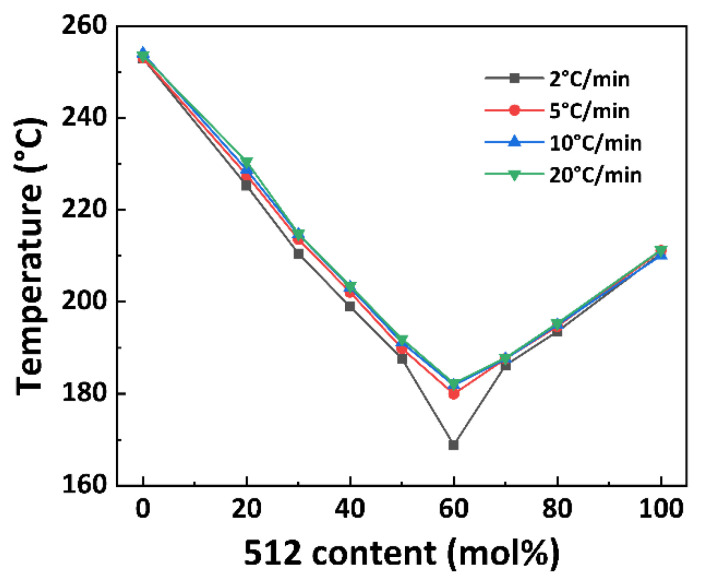
The melting temperature curve of PA56/512 with different contents of 512.

**Figure 8 polymers-15-02345-f008:**
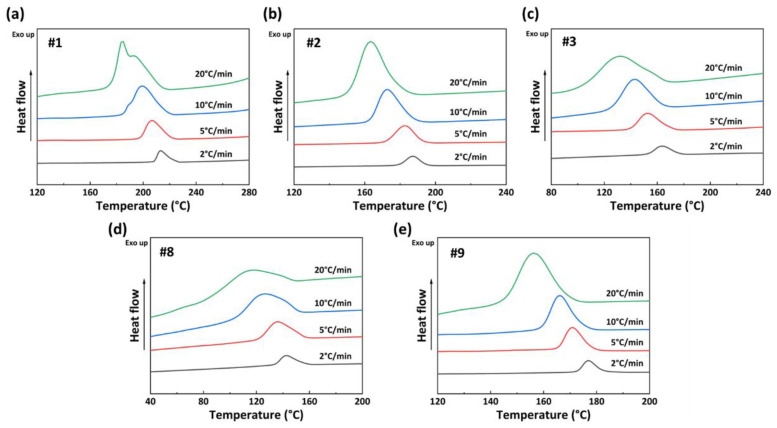
Non-isothermal crystallization curves of PA56/512 at different cooling rates: (**a**) #1, (**b**) #2, (**c**) #3, (**d**) #8, (**e**) #9.

**Figure 9 polymers-15-02345-f009:**
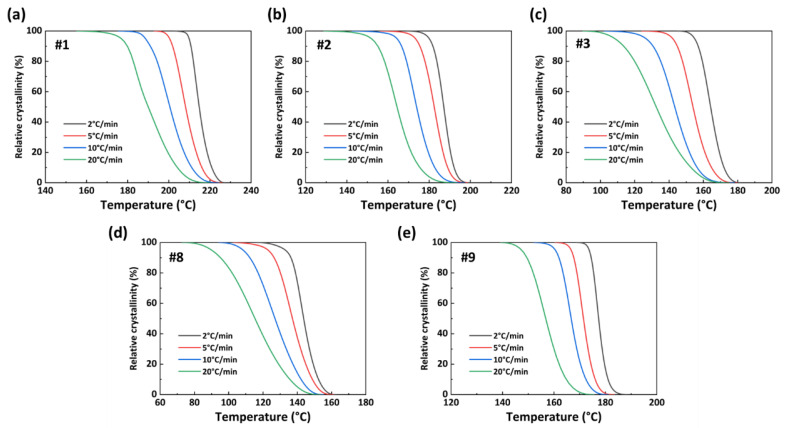
*X(T)*−*T* curves of PA56/512: (**a**) #1, (**b**) #2, (**c**) #3, (**d**) #8, (**e**) #9.

**Figure 10 polymers-15-02345-f010:**
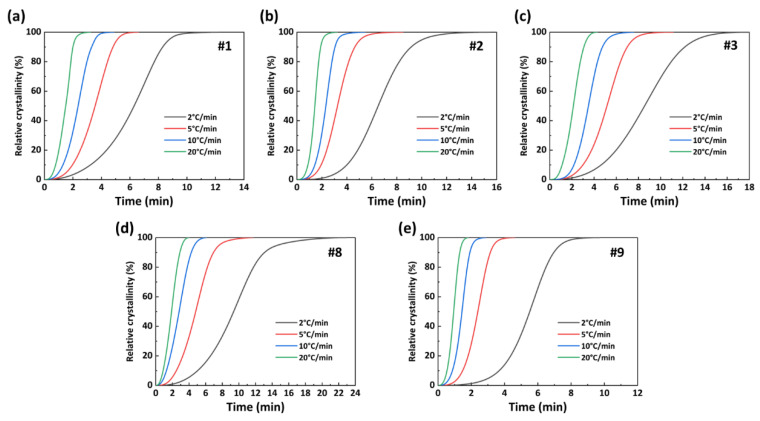
*X(t)*−*t* curves of PA56/512: (**a**) #1, (**b**) #2, (**c**) #3, (**d**) #8, (**e**) #9.

**Figure 11 polymers-15-02345-f011:**
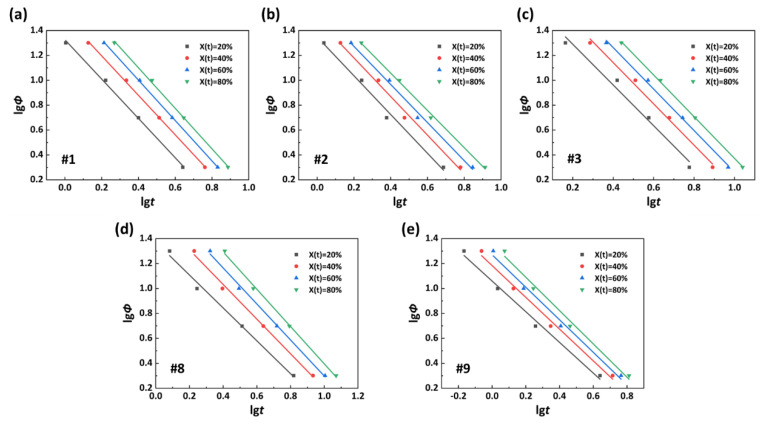
lgΦ−lg*t* curves of PA56/512: (**a**) #1, (**b**) #2, (**c**) #3, (**d**) #8, (**e**) #9.

**Figure 12 polymers-15-02345-f012:**
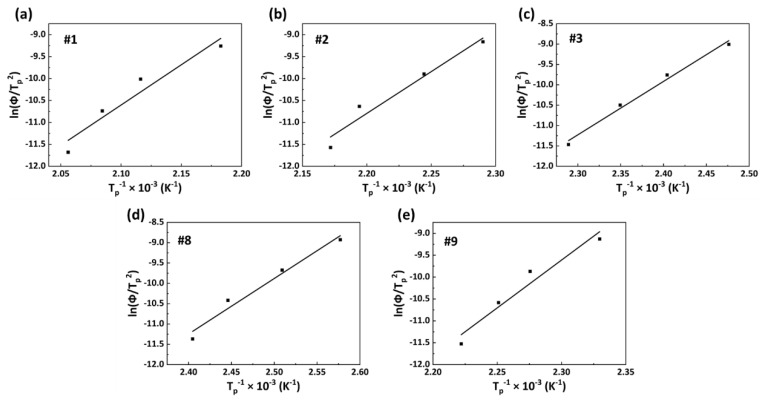
ln(Φ*/T_p_^2^*)−*T_p_*^−1^ curves of PA56/512: (**a**) #1, (**b**) #2, (**c**) #3, (**d**) #8, (**e**) #9.

**Table 1 polymers-15-02345-t001:** The ratios of integral data in each region of the ^1^H NMR spectra and other relevant values for PA56/512.

Sample	S_#a_:S_#b_:S_#c_:S_#d_		Relative Viscosity	[NH_2_] (mol/t)	*M*_n_−[NH_2_] (g/mol)
	Theoretical Value	Experimental Value			
#1	1:1:2:0.5	1:1:2:0.53	2.89	38.4	26,041.7
#2	1:1:2:1.1	1:1:2:1.04	2.82	41.3	24,213.1
#3	1:1:2:1.4	1:1:2:1.33	2.71	45.7	21,881.8
#4	1:1:2:1.7	1:1:2:1.63	2.69	46.4	21,551.7
#5	1:1:2:2.0	1:1:2:1.94	2.60	48.2	20,746.9
#6	1:1:2:2.3	1:1:2:2.25	2.53	51.2	19,531.3
#7	1:1:2:2.6	1:1:2:2.59	2.59	48.5	20,618.6
#8	1:1:2:2.9	1:1:2:2.82	2.55	49.1	20,366.6
#9	1:1:2:3.5	1:1:2:3.54	2.46	50.4	19,841.3

**Table 2 polymers-15-02345-t002:** Thermal decomposition data of PA56/512 with different composition ratios from TGA.

Sample	*T*_D_/°C	*T*_20%_/°C	*T*_M_/°C
#1	277.1	416.5	444.6
#2	281.8	416.7	445.6
#3	284.8	416.9	450.4
#4	297.6	422.6	454.9
#5	299.2	428.2	460.4
#6	309.4	429.7	463.6
#7	311.4	436.4	464.7
#8	318.8	440.1	466.2
#9	316.4	441.6	462.6

**Table 3 polymers-15-02345-t003:** Non-isothermal crystallization kinetic parameters of PA56/512 with different composition ratios.

Sample	Φ (°C/min)	*T_0_* (°C)	*T_p_* (°C)	*T_e_* (°C)	Δ*T_c_* (°C)	Δ*H_c_* (J/g)	*t*_1/2_ (min)
#1	2	227.4	213.2	203.0	24.4	50.0	6.30
	5	225.9	206.6	193.0	32.9	50.0	3.56
	10	224.1	199.4	175.5	48.6	54.3	2.36
	20	219.8	185.0	155.3	64.5	49.4	1.50
#2	2	200.1	187.3	169.3	30.8	44.6	6.52
	5	198.7	182.6	156.2	42.5	43.5	3.25
	10	197.1	172.4	142.7	54.4	49.5	2.32
	20	193.1	163.5	128.9	64.2	49.9	1.44
#3	2	181.3	163.7	145.4	35.9	36.3	8.58
	5	179.7	152.5	124.1	55.6	37.1	5.17
	10	178.2	142.8	102.4	75.8	44.0	3.48
	20	175.1	130.7	89.8	85.3	33.8	2.13
#8	2	162.5	142.7	116.6	45.9	55.6	9.37
	5	161.2	135.7	102.7	58.5	60.5	4.81
	10	155.0	125.4	94.1	60.9	55.1	2.81
	20	153.3	114.9	72.8	80.5	33.6	1.90
#9	2	188.4	176.9	169.0	19.4	53.9	5.51
	5	183.4	171.1	160.4	23.0	54.9	2.39
	10	181.1	166.3	152.3	28.8	53.5	1.44
	20	175.6	156.1	139.2	36.4	57.9	0.94

**Table 4 polymers-15-02345-t004:** Non-isothermal crystallization kinetic parameters of PA56/512 based on the Mo method.

Sample	Parameter	*X(t)* = 20%	*X(t)* = 40%	*X(t)* = 60%	*X(t)* = 80%
#1	*F(T)*	21.18	32.64	44.82	56.52
	*a*	1.58	1.59	1.62	1.63
#2	*F(T)*	22.32	30.98	38.00	45.45
	*a*	1.56	1.55	1.53	1.50
#3	*F(T)*	41.45	64.06	85.15	111.98
	*a*	1.64	1.66	1.67	1.68
#8	*F(T)*	23.56	38.77	55.34	77.90
	*a*	1.32	1.39	1.45	1.50
#9	*F(T)*	11.47	15.26	18.64	22.76
	*a*	1.23	1.27	1.30	1.34

**Table 5 polymers-15-02345-t005:** The crystallization activation energy of PA56/512.

Sample	#1	#2	#3	#8	#9
Δ*E* (kJ/mol)	156.6	154.2	109.1	113.6	181.5

## Data Availability

The following are available online in section “MDPI Research Data Policies” at https://www.mdpi.com/ethics (accessed on 10 May 2023).

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
