# Peer review of "The Investigation of Copolymer Composition Sequence on Non-Isothermal Crystallization Kinetics of Bio-Based Polyamide 56/512"

_polymers, 2023, doi:10.3390/polym15102345_

Round 1
Reviewer 1 Report
The authors conducted a series of experiments including the synthesis of polyamide 56/512 copolymers, and characterization via FTIR, NMR, TGA, and DSA to investigate the structure and physical and thermal properties. They have shown that PA56/512 has non-isothermal crystallization. Although they have done a throughout study and characterization of the resulting polymers, the novelty of this bio-based PA remains questionable. Thus, the reviewer recommends publishing this paper after the following minor question is resolved.
As one of the reasons, the author pointed out that pure PA56 has poor toughness and high-water absorption, affecting mechanical, electrical, and stability. Since the authors' strategy to make improvement is through copolymerized PA56 with PA512, what is the enhancement of performance of the product compared to pure PA56 or pure 512 needs to be clearly stated.
Author Response
Thank you for your careful suggestion.The reply as attecments.

Reviewer 2 Report
Dear Authors and Editor,
I read the paper carefully, and some issues must be clarified:
1. Please define PDA, AA and DDP.
2. Explain the structure of PA512.
3. Presents the novelty of the paper.
4. In Table 1, modify the units for [NH2] concentration.
5. ‘’The better thermal stability of pure PA512 compared with pure PA56, resulting from the lower hydrogen bond density of PA512 than that of PA56’. Please explain.
6. The English must be checked.
7. Page 9, modify PA5T/56.
8. Please modify the caption of Figure 8.
9. Why the average crystallization rate of pure PA512 is better than that of pure PA56?
English must be polished.
Author Response

(The authors gave the same response as above.)

Round 2
Reviewer 2 Report
Accept
Minor